# Contributions of treatment centre and patient characteristics to patient-reported experience of haemodialysis: a national cross-sectional study

Janine Hawkins [ORCID],[1] Nigel Smeeton [ORCID],[2] Amanda Busby,[1] David Wellsted,[1] Beth Rider,[1] Julia Jones [ORCID],[2] Retha Steenkamp,[3] Catherine Stannard,[3] Rachel Gair,[3] Sabine N van der Veer,[4] Claire Corps,[5] Ken Farrington[1,6]

JH and NS are joint first authors.

¹Life and Medical Sciences, University of Hertfordshire, Hatfield, UK
²Health and Social Work, University of Hertfordshire, Hatfield, UK
³UK Renal Registry, Renal Association, Bristol, UK
⁴Centre for Health Informatics, The University of Manchester, Manchester, UK
⁵St James's University Teaching Hospital, Leeds Teaching Hospitals NHS Trust, Leeds, UK
⁶Renal Unit, Lister Hospital, Stevenage, UK

**Correspondence to**
Ms Amanda Busby;
a.busby@herts.ac.uk

## ABSTRACT

**Objectives** To examine the relative importance of patient and centre level factors in determining self-reported experience of care in patients with advanced kidney disease treated by maintenance haemodialysis (HD).

**Design** Analysis of data from a cross sectional national survey; the UK Renal Registry (UKRR) national Kidney patient-reported experience measure (PREM) survey (2018). Centre-level data were obtained from the UKRR report (2018).

**Setting** National survey of patients with advanced kidney disease receiving treatment with maintenance HD in UK renal centres in 2018.

**Participants** The Kidney PREM was distributed to all UK renal centres by the UKRR in May 2018. Each centre invited patients receiving outpatient treatment for kidney disease to complete the PREM. These included patients with chronic kidney disease, those receiving dialysis—both HD and peritoneal dialysis, and those with a functioning kidney transplant. There were no formal inclusion/exclusion criteria.

**Main outcome measures** The Kidney PREM has 38 questions in 13 subscales. Responses were captured using a 7-point Likert scale (*never* 1, *always* 7). The primary outcome of interest was the mean PREM score calculated across all questions. Multilevel modelling was used to determine the proportion of variation of the mean PREM score across centres due to patient-related and centre-related factors.

**Results** There were records for 8253 HD patients (61% men, 77% white) from 69 renal centres (9–710 patients per centre). There was significant variation in mean PREM score across centres (5.35–6.53). In the multivariable analysis there was some variation in relation to both patient- and centre-level factors but these contributed little to explaining the overall variation. However, multilevel modelling showed that the overwhelming proportion of the explained variance (45%) was explained by variation between centres (40%), only a small proportion of which is identified by measured factors. Only 5% of the variation was related to patient-level factors.

**Conclusions** Centre rather than patient characteristics determine the experience of care of patients receiving HD. Further work is required to define the characteristics of the treating centre which determine patient experience.

### Strengths and limitations of this study

⇒ A large sample size deriving from a national study involving all but two UK renal centres.
⇒ We employ multilevel analysis to explore the factors that inform patient experience of renal care at both patient-level and centre-level and provide accurate estimates of their relative contributions.
⇒ Patient-reported experience measure results do not take into account the potential impact of negative non-response bias, that is, the tendency to participate in the survey being negatively associated with problem reporting.
⇒ While distribution of patient scores is skewed, sensitivity modelling indicated that the reported models and interpretation are robust.

## INTRODUCTION

As the survival of patients improves, attention is shifting from clinical and biomedical 'hard metrics' as primary indicators of healthcare quality[1] towards gathering feedback from patients on aspects of their experience consistent with their values; including health, well-being, respect and dignity, quality of life (QoL) and independence.[2] Patient experience of care, defined as 'patients' perception of the range of interactions they have with the healthcare system they use'[3–5] is increasingly recognised as an indicator of healthcare quality.[5–7] The patient perspective has traditionally been assessed through measures of patient satisfaction.[2] Patient-reported experience measures (PREMs) are emerging as a favoured metric for assessing healthcare quality.[3 7 8] PREMs are standardised self-report tools designed to tap into specific domains of care (ie, communication, care environment, healthcare team). Unlike satisfaction measures, patient-reported experience captures patient perspective of quality

by assessing the extent to which specific care processes occur over time[3 9] (eg, 'In the last 12 months how often did your doctor give you the information you needed?'). Patient experience allows providers to monitor the extent to which fundamental standards of healthcare quality have been met, through objective reporting.[3 7] In contrast, patient satisfaction measures assess the extent to which patients are *happy* with care, which is a reflection of patients' own subjective standards, expectations and affect, which is highly influenced by patient gratitude for the care received, and provides a less reliable measure of healthcare quality.[3 7 10] PREMs aim to minimise subjective bias that arises from patient satisfaction measures.[6] Nevertheless, patient experience influences patient satisfaction, demonstrating a causal link between these quality metrics.[10] Measuring patient experience offers a unique understanding of how we appraise healthcare quality, by responding to the voice of those at the centre of care.[7] Evaluated alongside patient safety and effectiveness, patient experience contributes to an overall holistic representation of healthcare quality.[6 7 11]

PREMs can be used by healthcare providers, policymakers and regulators to assess provider performance and facilitate quality improvement initiatives.[1] PREMs are also used to inform pay-for-performance schemes, by which services may receive funding increments in line with their performance.[8 9] Survey results allow for comparisons across providers[1] and can be made publicly available for use by patients and the general public. Patient experience is linked to a range of outcomes across healthcare populations; including clinical effectiveness, self-rated and objectively measured health outcomes, patient activation and treatment adherence.[9 12] This demonstrates the clinical utility of measuring patient-reported experience towards optimising healthcare delivery and patient outcomes. In the UK, approximately 6% of the adult population have a diagnosis of chronic kidney disease (CKD). Most have mild to moderate disease (stages 1–3), with around 0.4% having advanced CKD (stages 4 and 5). Renal replacement therapy (RRT) is required for many patients with stage 5 disease. Around 60 000 patients in the UK receive RRT, approximately half of whom are living with a renal transplant and the remainder receiving dialysis.[13] Haemodialysis (HD) is the most prevalent dialysis modality. HD patients require long-term and burdensome treatment throughout their lifetime.[6] The quality of this care can have a significant impact on these patients' wellbeing, health and QoL.[6 14] Health-related QoL is often indicated by patients as being more important to them than survival.[15]

Introduced in 2016, the UK Renal Registry (UKRR) Kidney PREM (online supplemental file 1) is part of a national programme that collects annual feedback from patients with kidney disease on their experience of National Health Service (NHS) renal services, including those on HD. However, the implementation of PREMs in clinical practice has been relatively limited.[6] As a result, responding to patient experience feedback remains a challenge for providers.[16] To translate patient feedback into actionable information to drive quality improvements,[6] we must be able to define which factors inform patient experience. Whether patient experience is mainly determined by patient characteristics[17 18] or by those of the centre remains unknown.[19] There is a growing body of literature addressing patient experience of care (and the factors which inform it), but few studies acknowledge the hierarchical framework of patients nested within centres. No previous studies have taken account of this framework in the HD population. The objective of the current study was to examine the relative importance of patient- and centre- level factors in determining self-reported experience of care in patients with advanced kidney disease treated by maintenance HD. We used clustered analysis (multilevel regression models) to examine the association between patient demographic and centre-level factors and patient experience, using secondary data from the national 2018 Kidney PREM survey. We have focused on patients receiving HD to reduce the heterogeneity of the sample and because data suggest poorer experience of care in this group.[20] By using analytical methods that account for grouping within treatment centre, our study aims to provide evidence that treating centre makes a larger contribution to patient experience than patient characteristics, thereby supporting a systems perspective towards understanding patient experience.[1]

## METHODS

### Kidney PREM data

The validated Kidney PREM questionnaire was developed by a group of patients, clinicians and academics as a disease specific measure of patient experience. The third survey, conducted in 2018,[21] involved 71 adult renal centres across the UK, with kidney patients given the option of a paper version of the questionnaire, distributed and collected by centres, or completion online. Patient age, gender and ethnicity were recorded along with current treatment (peritoneal dialysis; HD and its location (at home, in hospital, in satellite); transplanted or attending kidney clinics but not on dialysis or in receipt of a transplant). Patients reported their use of hospital transport by answering transport-specific questions. Information regarding patient experience of care was collected from 38 questions grouped into 13 subscales consisting of between two and five questions, with a final question asking the patient to score their overall experience of the service provided. Responses were captured using a 7-point Likert scale from *never (1) to always (7),* with additional options of *don't know* and *not applicable..* The research reported here investigated only those patients receiving HD. The sample size was determined by the number of HD patients providing a completed Kidney PREM questionnaire in the 2018 survey.

## Centre-level population characteristics

In addition to the patient-level survey data, the UKRR provides centre-level population characteristics (centre-level data). UKRR has collected centre-based data and published annual reports since 1998. Information provided for this study was from 2017,[13] the most recently available. Centre demographic variables included mean patient age, percentage male, ethnicity distribution (Asian, Black, White, Other, Rather not say) and mean index of multiple deprivation. Follow-up information was obtained on the percentage of patients alive at the end of 2016 that died during 2017. Other centre-level information related to dialysis access for patients receiving RRT, current or previous smoking, comorbidities at the start of RRT (eg, angina, previous myocardial infarction, diabetes), the percentage of patients who were late presenters (ie, less than 90 days between first nephrology appointment and commencing RRT), and mean systolic and diastolic blood pressures. Mean blood test levels were provided for phosphate, bicarbonate, potassium, ferritin and urea reduction ratio. Information on transplant waiting lists and transplant rate was also obtained.

## Analysis

The primary outcome of interest was the mean PREM score calculated across all 38 questions across the 13 subscales, analysed as a continuous variable, with a difference of greater than 10% (0.7) of the scale range (1–7) being regarded as clinically important. Scores were calculated where patients had answered five or more of the 33 PREM questions which are not filtered for treatment type. Details of how the mean score was adjusted for missing responses is provided in the technical annex to the 2018 report.[21]

The information available from the completed PREM questionnaires was used to obtain the patient-level variables, these being: age in years (17–21, 22–30, 31–40, 41–55, 56–64, 65–74, 75–84, 85+), gender (male, female, rather not say), ethnicity (Asian, Black, White, Other, Rather not say), use of hospital transport (yes, no) and HD location (in-hospital, in-satellite, at home). Clinically relevant centre-level variables were selected. As the range of possible values differs between the constituent countries of the UK, for comparability, deciles for the Index of Multiple Deprivation were calculated. The strength of association between mean PREM score and prevalent mortality measures was assessed. These were the percentage of patients alive at the end of 2016 who died during 2017 (prevalent 1-year mortality unadjusted) and prevalent 1-year mortality adjusted to age 60 years.[13]

A multivariable linear regression, which assumes independence of observations, was performed on the patient-level and centre-level variables that were at least 85% complete and available for each centre. Variables with a p value of less than 0.1 were selected for multilevel modelling. Multilevel analyses were then performed on the mean scores by using the iterative generalised least squares method of estimation. Variance partition coefficients were calculated. These represent the proportion of total variation in an outcome that is due to differences occurring at each level; a high coefficient at centre-evel indicates that more of the variation in the model is due to differences between centres than between patients. First, the variance partition coefficients were calculated for the null model to give raw coefficients. Then a model was fitted adjusting for patient-level characteristics to give patient adjusted coefficients. Next, both patient-level and centre-level characteristics were entered. Finally, any interactions between variables identified were added in to give fully adjusted coefficients.

Sets of models were compared using likelihood ratio tests. In order to assess whether the patients entered into the analysis were similar to those excluded due to incomplete information, the two groups were compared on age group, gender, ethnicity and location of treatment.

Sensitivity analyses were performed as part of the modelling. As patient PREM scores were negatively skewed, an exponential transformation was applied to bring the observations closer to a normal distribution. Additionally, results were compared with and without the robust estimation of errors. Snijders-Bosker $R^2$ values were estimated after fitting the multilevel model to determine the proportion of variance in scores attributable to centre-level and patient-level differences.

All analyses were conducted using Stata V.IC/15.1.

## Patient and public involvement

The UKRR Patient Council was consulted regarding how best to share this research, as well as the Kidney PREM and other related research, with patients, medical professionals and the public in general. The study was presented at a Patient Council meeting in March 2019 and the members' views sought in particular on our dissemination plans to maximise the impact of the research on patient care.

Feedback from the Council was that there are many ways to engage with the kidney community such as through public user group fora, patient blogs or via the Kidney Patient Associations. It was also felt that it would be beneficial to send results directly to centre staff, as well as the hospital chief executive, medical director and clinical director to ensure the findings were received at all organisational levels. In addition, the Council were keen that treatment centres were sent their own results from the Kidney PREM, highlighting how they compare to other centres, to ensure that best practice was shared across settings.

## RESULTS

There were records for 8253 HD patients from 69 renal centres (9–710 patients per centre) with two centres providing no responses. Overall, reported experience of PREM was positive (mean=5.93, SD=1.00). There was noticeable variation between centres. Mean scores were normally distributed, ranging from 5.35 to 6.53 (median

**Table 1** Mean reported patient experience scores by patient characteristics in the analysed sample (percentages out of patients with the information present, n=8253)

| Variable | Observations | Mean | SD | Median |
|---|---|---|---|---|
| Age (years) | | | | |
| 17–21 | 28 (0.3%) | 5.71 | 1.04 | 5.94 |
| 22–30 | 160 (2.0%) | 5.79 | 1.07 | 6.05 |
| 31–40 | 325 (4.0%) | 5.87 | 1.10 | 6.28 |
| 41–55 | 1314 (16.1%) | 5.80 | 1.11 | 6.14 |
| 56–64 | 1540 (18.9%) | 5.86 | 1.06 | 6.15 |
| 65–74 | 2255 (27.7%) | 6.00 | 0.96 | 6.31 |
| 75–84 | 2046 (25.1%) | 6.01 | 0.91 | 6.27 |
| 85+ | 477 (5.9%) | 6.01 | 0.94 | 6.28 |
| Missing | 108 | | | |
| Gender | | | | |
| Male | 4481 (60.8%) | 5.96 | 0.98 | 6.26 |
| Female | 2861 (38.8%) | 5.91 | 1.00 | 6.21 |
| Not say | 33 (0.4%) | 4.43 | 1.42 | 4.36 |
| Missing | 878 | | | |
| Ethnicity | | | | |
| Asian | 893 (11.5%) | 5.74 | 1.05 | 5.97 |
| Black | 541 (6.9%) | 5.83 | 1.04 | 6.09 |
| White | 6025 (77.3%) | 5.98 | 0.97 | 6.28 |
| Other | 210 (2.7%) | 5.99 | 0.93 | 6.27 |
| Not say | 129 (1.7%) | 5.12 | 1.40 | 5.29 |
| Missing | 455 | | | |
| Location of treatment | | | | |
| Home | 270 (3.4%) | 6.12 | 0.91 | 6.42 |
| Satellite | 4131 (51.6%) | 5.93 | 1.00 | 6.24 |
| Hospital | 3600 (45.0%) | 5.93 | 1.01 | 6.22 |
| Missing | 252 | | | |
| Hospital transport | | | | |
| Yes | 5226 (63.3%) | 5.90 | 1.01 | 6.20 |
| No | 3027 (36.7%) | 5.98 | 0.99 | 6.29 |
| Missing | 0 | | | |

5.96, IQR 5.79–6.13). In addition, centres differed markedly in the proportion of patients giving a low experience score. Taking scores of less than 5 to be low, there was a 10-fold difference between centres (3.0%–33.3%). The range of PREM scores (1.18) is far greater than 10% (0.7) of the scale range (1–7) and more than 1 SD of the overall score, giving an indication of the impact of the centre-level variation. Table 1 shows PREM summary scores for groups based on demographic characteristics. Of note, those unwilling to disclose their gender or ethnicity (selecting 'rather not say') on average gave lower experience scores. Some variation was seen across age groups, with those over 65 years reporting a better experience. However, the difference was small, ranging from 5.71 (18–21 years) to 6.01 (5+ years).

Those centre-level variables with at least 85% completeness and recorded by all renal centres are shown in table 2. These consisted of sociodemographic variables at both patient summary and centre levels, centre size, transplanting centre (yes/no), blood test results and waiting list information. Although the data for percentage White at renal centre level were more than 85% complete, they were unavailable for seven centres, all in Scotland, and so were excluded. However, there was a strong negative correlation between percentage White and size of centre (r=–0.7922) suggesting that centre size is a good proxy variable for percentage White. No association was found between mean PREM score and prevalent mortality, either unadjusted (r=0.0482), or age-adjusted (r=−0.0500).

**Table 2** Descriptive statistics for centres

|  | N | Mean | SD | Range across centres |
|---|---|---|---|---|
| Mean age* (years) | 69 | 65.4 | 2.5 | 60.0–72.8 |
| Percentage male* | 69 | 62.0 | 4.1 | 47.1–70.3 |
| Mean Index of Multiple Deprivation Decile* | 69 | 4.7 | 1.1 | 1–7 |
| Transplanting centre* |  |  |  |  |
| Yes | 22 (31.9%) |  |  |  |
| No | 47 (68.1%) |  |  |  |
| Size of centre* | 69 | 891 | 685 | 124–3417 |
| Mean haemoglobin level (g/L)* | 69 | 109.2 | 2.6 | 103.4–116.0 |
| Mean phosphate level (mmol/L) | 69 | 1.54 | 0.13 | 0.79–1.74 |
| Mean ferritin (ng/mL) | 66 | 492 | 116 | 297–889 |
| Wait listed within 2 years starting RRT (%) | 69 | 40.9 | 11.6 | 14.3–65.4 |
| Transplant within 2 years of waiting list (%) | 69 | 53.4 | 15.3 | 25–100 |
| Prevalent 1-year mortality unadjusted (%) | 69 | 16.6 | 3.3 | 10.2–25.7 |
| Prevalent 1-year mortality adjusted to age 60 (%) | 69 | 12.0 | 2.3 | 6.4–17.3 |

* Selected for multilevel modelling
RRT, renal replacement therapy.

The results from the multivariable linear regression are shown in table 3. All patient-level variables were retained in the model, along with selected centre-level variables as indicated in table 2. Adjusting for other patient- and centre-level variables increased the effect of age on the PREM scores, with a wider range of 0.53 across the age groups (compared with 0.3 as shown in table 1). Several centre-level variables were found to be reliably associated with mean scores in the adjusted model, however, the range of effect for any variable was limited (95% CI 0.03 to 0.44).

Table 4 provides details of relationships between the explanatory variables at the level of centre and patient, and the mean patient experience score. Data for the multilevel modelling were obtained from 6945 (84.2%) patients, with 69 centres represented. Patients who were included differed from those excluded in terms of ethnicity (White: 77.8% vs 72.6%) and location of treatment (hospital: 45.4% vs 42.0%). However, age (65+: 58.8% vs 57.9%), gender (male: 60.8% vs 59.5%) and use of hospital transport (yes: 62.9% vs 65.8%) were similar.

At the centre level, higher mean scores were associated with a higher percentage placed on the waiting list for a transplant within two years of starting RRT. Lower scores were associated with larger centres. At the patient level, lower mean scores were associated with younger patients (<56 years), refusal to disclose gender, refusal to disclose ethnicity, use of hospital transport and receipt of HD in-centre or in-satellite (compared with dialysis at home). Most of these effects were relatively small (<0.4 on a scale from 1 to 7), the main exceptions being refusal to disclose gender and refusal to disclose ethnicity. Patient-level variables had a similar effect on scores in the multilevel model as they did in the multivariable linear

regression model (tables 3 and 4). The sensitivity analyses performed by firstly dropping robust estimation and then using the transformed data each produced similar results to those of the main analysis.

Table 5 shows the estimated variance for each model. The total variance decreases as the patient-level variables are added and decreases further once adjusted for centre-level variables (ie, as more of the variance in patient scores was explained). The Snijders-Bosker $R^2$ values[22] suggest that 40.1% of variation in scores is explained by centre variation and just 5.3% of variation is accounted for by differences in patient characteristics.

## DISCUSSION
### Statement of principal findings
This study provides important insights into what influences patient experience of care in HD patients in this national sample of respondents of the UKRR/KCUK Kidney PREM 2018 survey. In this study, patient characteristics—younger patients (<56 years), refusal to disclose gender, refusal to disclose ethnicity, use of hospital transport and receipt of treatment in centre or satellite—were only modestly associated with patient experience. These effects are small in magnitude, indicating a limited impact on patient experience. Collectively, patient characteristics account for just 5% of the observed variance in the PREM score. The only important effects were observed consistently in those patients who refused to disclose their gender or ethnicity. Previous studies have also described small variations in ratings for self-reported patient experience among older patients, those experiencing shortened treatments[18] and some minority ethnic populations.[19]

**Table 3** Results for the multivariable linear regression. (n=6795, 69 centres)

| Variable | Coefficient | 95% CI | P value |
|---|---|---|---|
| Patient-level | | | |
| Age (baseline 56–64) | | | |
| 17–21 | −0.334 | (−0.734 to 0.067) | 0.102 |
| 22–30 | −0.058 | (−0.233 to 0.117) | 0.515 |
| 31–40 | 0.028 | (−0.101 to 0.156) | 0.673 |
| 41–55 | −0.032 | (−0.111 to 0.047) | 0.424 |
| 56–64 | – | – | – |
| 65–74 | 0.119 | (0.050 to 0.187) | 0.001 |
| 75–84 | 0.146 | (0.075 to 0.217) | <0.001 |
| 85+ | 0.197 | (0.087 to 0.307) | <0.001 |
| Gender (baseline male) | | | |
| Female | −0.030 | (−0.078 to 0.018) | 0.223 |
| Not say | −1.072 | (−1.468 to −0.677) | <0.001 |
| Ethnicity (baseline White) | | | |
| Asian | −0.054 | (−0.135 to 0.027) | 0.193 |
| Black | −0.078 | (−0.176 to 0.020) | 0.121 |
| Other | 0.125 | (−0.019 to 0.270) | 0.089 |
| Not say | −0.627 | (−0.826 to −0.429) | <0.001 |
| Location of treatment (baseline home) | | | |
| Satellite | −0.149 | (−0.282 to −0.016) | 0.028 |
| Hospital | −0.219 | (−0.353 to −0.086) | 0.001 |
| Hospital transport | −0.120 | (−0.170 to −0.070) | <0.001 |
| Centre-level | | | |
| Mean age (per year) | 0.024 | (0.009 to 0.039) | 0.002 |
| Percentage male (per 10%) | −0.110 | (−0.185 to −0.036) | 0.004 |
| Mean Index of Multiple Deprivation (IMD) decile | −0.057 | (−0.086 to −0.027) | <0.001 |
| Transplant centre | 0.274 | (0.200 to 0.348) | <0.001 |
| Centre size (per 100 patients) | −0.024 | (−0.030 to −0.018) | <0.001 |
| Mean haemoglobin (g/L) | 0.013 | (0.002 0.024) | 0.026 |
| Mean phosphate (mmol/L) | 0.022 | (−0.197 to 0.240) | 0.846 |
| Mean ferritin (ng/mL) | −0.0001 | (−0.0003 to 0.0001) | 0.297 |
| On waiting list <2 years from starting RRT (per 10%) | 0.075 | (0.049 to 0.102) | <0.001 |
| Transplant within 2 years of waiting list (per 10%) | 0.015 | (−0.007 to 0.037) | 0.184 |
| Unadjusted prevalent mortality (per 10%) | −0.016 | (−0.108 to 0.076) | 0.736 |

RRT, renal replacement therapy.

In contrast, the centre accounts for an overwhelming proportion of the variance (40%). However, our findings indicate that while centre plays a major role in determining patient experience of care, the centre-level factors that drive patient experience are unknown. Any associations between collected centre factors and patient experience were modest and of doubtful clinical importance (ie, <10% (0.7) of the 1–7 scale range) and likely to have limited impact on patient experience. For example, lower scores were associated with larger centres (−0.027 per 100 patients, p<0.001) and higher scores associated with a higher percentage of patients placed on the transplant waiting list within 2 years of starting RRT (0.055 per 10%, p=0.007). These findings challenge the concept that providing care focused on patient characteristics is the key determinant of patient experience. In keeping with this, Brady et al[19] have reported variation in mean patient experience scores between groups of US facilities with different characteristics, for example, size of facility, hospital-based versus free-standing, for-profit versus non-profit. Differences were relatively small, yet still outweighed those attributed to patient characteristics. Factors at a group (cluster) level (eg, hospitals

**Table 4** Results for the multilevel modelling. (n=6945, 69 centres)

| Variable | Coefficient | 95% CI | P value |
|---|---|---|---|
| Patient-level | | | |
| Age (baseline 56–64) | | | |
| 17–21 | −0.359 | (−0.806 to 0.089) | 0.116 |
| 22–30 | −0.088 | (−0.287 to 0.112) | 0.389 |
| 31–40 | 0.022 | (−0.098 to 0.142) | 0.715 |
| 41–55 | −0.035 | (−0.116 to 0.046) | 0.396 |
| 65–74 | 0.118 | (0.047 to 0.188) | 0.001 |
| 75–84 | 0.147 | (0.083 to 0.212) | <0.001 |
| 85+ | 0.219 | (0.124 to 0.313) | <0.001 |
| Gender (baseline male) | | | |
| Female | −0.027 | (−0.076 to 0.022) | 0.275 |
| Not say | −1.074 | (−1.662 to −0.485) | <0.001 |
| Ethnicity (baseline white) | | | |
| Asian | −0.054 | (−0.166 to 0.057) | 0.339 |
| Black | −0.052 | (−0.162 to 0.058) | 0.352 |
| Other | 0.122 | (−0.059 to 0.304) | 0.186 |
| Not say | −0.625 | (−0.916 to −0.334) | <0.001 |
| Location of treatment (baseline home) | | | |
| Satellite | −0.127 | (−0.258 to 0.005) | 0.060 |
| Hospital | −0.231 | (−0.383 to −0.079) | 0.003 |
| Hospital transport | −0.128 | (−0.183 to −0.072) | <0.001 |
| Centre-level | | | |
| Mean age (per year) | 0.015 | (−0.012 to 0.043) | 0.268 |
| Percentage male (per 10%) | −0.122 | (−0.232 to −0.011) | 0.031 |
| Mean IMD decile | −0.035 | (-0.086 to 0.016) | 0.174 |
| Transplant centre | 0.291 | (0.165 to 0.416) | <0.001 |
| Centre size (per 100 patients) | −0.027 | (−0.037 to −0.018) | <0.001 |
| Mean haemoglobin (g/L) | 0.011 | (−0.009 to 0.031) | 0.267 |
| On waiting list <2 years from starting RRT (per 10%) | 0.055 | (0.015 to 0.095) | 0.007 |

RRT, renal replacement therapy.

**Table 5** Variance at level of centre and patient, before and after adjustment for explanatory variables

| Level of adjustment | Variance (total) | Variance (cons) | Variance (residual) | Variance accounted for | |
|---|---|---|---|---|---|
| | | | | Patient level variables (%)[*] | Centre level variables (%)[*] |
| Unadjusted | 1.0166 | 0.0407 | 0.9759 | | |
| Adjusted for patient level variables | 0.9763 | 0.0433 | 0.9330 | | |
| Adjusted for patient and centre level variables | 0.9486 | 0.0187 | 0.9299 | 5.3 | 40.1 |

*Snijders-Bosker R Squared

or GP practices) can have an influence on an individual's outcome of interest. This is typically handled in analysis using a group variable, or multilevel models. In sample size estimation where no current estimate of the 'cluster effect' is published it is generally accepted that a reasonable conservative estimate of the intra-cluster correlation is in the order of r=0.05.[23] Often, the effect of clustering is smaller. In this paper however we present evidence that indicates that for patient experience the variation between clusters (eg, treating centres, ie, the centre effect) is larger by several orders of magnitude. In fact, variation attributable to the observed patient-level variables (age, ethnicity and so on) is approximately 5%, while the variation attributable to the centre effect is approximately 40%.

### Strengths and limitations of the study

Our study has considerable strengths, most notably the large sample size deriving from a national study involving all but two of the UK renal centres making it more likely that the sample was representative, raising statistical power and hence the strength of the findings. The study is also the first to employ multilevel analysis to explore the factors that inform patient experience of renal care at both patient-level and centre-level and provide accurate estimates of the relative contributions for factors at both these levels. It has been suggested that centre-level numbers in excess of 50 are required to prevent bias in estimates of centre-level SEs.[24] It is worth noting that the very large sample size means that a number of so-called statistically significant effects (p values <0.05) represent small differences unlikely to be of clinical importance.

The study also has some limitations. Though PREM scores were generally high, the results do not take into account the potential impact of negative non-response bias, that is, the tendency to participate in the survey being negatively associated with problem reporting.[25] Furthermore, the distribution of scores is skewed, but sensitivity modelling indicated that the reported models and interpretation are robust. More importantly, the observed variation between centres was largely unmeasured. While there have been some calls to focus on variation between centres, there is as yet little effort to understand what these factors are. Based on our study findings, the need to address this gap is urgent.

### Meaning of the study: possible explanations and implications for clinicians and policymakers

These findings have implications for care delivery, suggesting that the benefits on patient experience of personalising care delivery according to patient characteristics may be far outweighed by efforts focused on reducing variation between centres as suggested by others.[1 26] However, it is unclear just what the centre characteristics are which drive this variation. Certainly, the factors examined in this study do not provide an adequate explanation. Wider systemic problems such as understaffing, lack of managerial support, and increased pressure to meet ambitious national targets can promote

a culture of care delivery which is task- rather than patient-focused.[27] Though these issues reflect challenges faced universally by the NHS, the centre-wide variation in patient experience observed in this study may indicate that some centres are better equipped than others to cope with increasing pressures, while others may lack the culture and mechanism to deliver the same standard of care.[1]

### Unanswered questions and future research

Further study is needed to determine those centre characteristics which most influence the experience of care reported by the UK HD population. The recognition of the need for 'deep cultural change'[28] in promoting quality improvement is consistent with these interpretations. Additionally, further study is needed to ascertain if this is the case across other service provisions, not just in renal settings.

In summary, this large national study of patients receiving HD suggests that it is centre rather than patient characteristics which are important in shaping patient experience of care. Our findings suggest that a shift in focus from patient characteristics to the service itself is required to understand and improve patient experience of HD and potentially of many other areas of healthcare. This study provides a clear direction for future research aimed at determining the nature of the centre characteristics which drive patient experience.

**Correction notice** This article has been corrected since it was published online. The corresponding author has been changed from "Janine Hawkins' to "Amanda Busby".

**Acknowledgements** Thank you to all the patients who completed the 2018 Kidney PREM, and the UK renal centres for providing data to the UK Renal Registry. We are grateful to the UK Renal Registry for supplying the data from the 2018 PREM and 2017 UK Renal Registry Report for analysis for this study, and to the UK Renal Registry Patient Council and for Kidney Care UK for their input.

**Contributors** Study design: KF, JH, JJ, NS, DW. PPI: CC, JH, JJ. Aggregation of data: AB, JH, NS, RS. Analysis of data: AB, KF, NS, SNvdV, DW. Interpretation of data/results: AB, CC, KF, RG, JH, JJ, BR, CS, NS, SNvdV, DW. Drafting manuscript: AB, KF, JH, JJ, BR, NS, SNvdV, DW. Approval of final manuscript: all authors.

**Funding** This project was funded internally by the University of Hertfordshire with a cross-school collaboration research grant.

**Competing interests** All authors have completed the ICMJE uniform disclosure form at www.icmje.org/coi_disclosure.pdf and declare: no support from any organisation for the submitted work; no financial relationships with any organisations that might have an interest in the submitted work in the previous three years; no other relationships or activities that could appear to have influenced the submitted work.

**Patient consent for publication** Not required.

**Ethics approval** The study involved secondary analysis of robustly anonymised UK Renal Registry data. The UK Renal Registry has NHS Health Research Authority research ethics approval (16/NE/0042).

**Provenance and peer review** Not commissioned; externally peer reviewed.

**Data availability statement** Data may be obtained from a third party. Data are held by the UK Renal Registry. To apply to access Registry data see https://www.renalreg.org/about-us/working-with-us/.

**Author note** Guarantor: Janine Hawkins.

**ORCID iDs**
Janine Hawkins http://orcid.org/0000-0002-7646-2821
Nigel Smeeton http://orcid.org/0000-0001-9460-5411
Julia Jones http://orcid.org/0000-0003-3221-7362

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
