## [Reviewer comments · BMJ Open]

ARTICLE DETAILS

TITLE (PROVISIONAL)	Contributions of treatment centre and patient characteristics to patient-reported experience of haemodialysis: A national cross-sectional study
AUTHORS	Hawkins, Janine; Smeeton, Nigel; Busby, Amanda; Wellsted, David; Rider, Beth; Jones, Julia; Steenkamp, Retha; Stannard, Catherine; Gair, Rachel; van der Veer, Sabine N.; Corps, Claire; Farrington, Ken

VERSION 1 – REVIEW

REVIEWER	fredric finkelstein Yale University USA
REVIEW RETURNED	03-Nov-2020

GENERAL COMMENTS	This is an interesting and well written paper. It would be helpful to see more detail on the range of facility scores. Mean scores varied between 5.35 and 6.53 -- which does not seem to me to represent a dramatic difference. What was the distribution of scores?? It would also be interesting to see if there is a relationship between these scores and standardized mortality rates or other objective measures of quality care (such as Hb levels,, etc).
---

REVIEWER	Karine Manera The University of Sydney
REVIEW RETURNED	20-Dec-2020

GENERAL COMMENTS	Hawkins et al. present a cross-sectional study examining the contribution of patient vs centre-level factors to patient-reported experience measure scores. This is a very well presented study with robust methods. Only minor comments: - As acknowledged by the authors, this study does leave the reader asking more questions than it answered. The authors could add to their discussion by comparing it to other related literature looking at the patient experience and patient/centre characteristics (e.g. Dad et al. 2018 BMC Nephrology) - Tables 3 and 4 need footnotes explaining the abbreviations (IMD).
--

VERSION 1 – AUTHOR RESPONSE

Reviewer 1

It would be helpful to see more detail on the range of facility scores. Mean scores varied between 5.35 and 6.53 -- which does not seem to me to represent a dramatic difference. What was the distribution of scores?

Centre mean PREM scores (mean = 5.93, standard deviation = 1.00) appeared approximately normally distributed (median 5.96, interquartile range 5.79-6.13). Although all mean centre scores indicated a relatively positive experience (5.35 - 6.53 from a maximum of 7), there was significantly more variation across centres when compared to variation across patient characteristics. As noted in the manuscript, we think this difference is large because "This range (1.18) is far greater than 10% (0.7) of the scale range (1 to 7) and more than 1 standard deviation of the overall score, giving an indication of the impact of the centre-level variation". To emphasise the point we have looked at the variation across centres in the proportion of patients with low (<5) scores. This demonstrated a greater than 10 fold variation across centres.

We have amended part of the first paragraph of Results to read:

"There was noticeable variation between centres. Mean scores were normally distributed, ranging from 5.35 to 6.53 (median 5.96, interquartile range 5.79-6.13). In addition, centres differed markedly in the proportion of patients giving a low experience score. Taking scores of less than 5 to be low, there was a ten-fold difference between centres (3.0% to 33.3%)".

It would also be interesting to see if there is a relationship between these scores and standardized mortality rates or other objective measures of quality care (such as Hb levels,, etc).

We agree wholeheartedly; our analysis was limited by the data available. We would ideally investigate patient reported outcome measures and prevailing culture as indicators of care quality, as well as measures we have managed to include, ie, haemoglobin, phosphate, URR and waiting times. The relationship between centre mean haemoglobin level levels and PREM scores was examined in the original version of the paper (see "Mean Haemoglobin Level (g/L) in tables 2 and 3). The haemoglobin level was associated with mean PREM scores in isolation (table 2) but no relationship remained when adjusted for other covariates.

We had considered mortality in the original version of the paper. There was a labelling error in both Tables 2 and 3, % deaths during the first 90 days should have been labelled "Prevalent 1 year mortality unadjusted" which refers to the percentage of patients alive at the end of 2016 who died during 2017. This is now labelled correctly in the revised Tables. We have also added a variable "Prevalent 1 year mortality adjusted to age 60" to Table 2. Neither was associated with mean PREM score in univariable regression. Unadjusted prevalent mortality was also included in multivariable linear model which demonstrated no independent relationship with mean PREM score (Table 3).

To clarify this for the reader, we have made a number of amendments to the paper.

Paragraph two under Methods, Analysis now reads:

"The information available from the completed PREM questionnaires was used to obtain the patient-level variables, these being: age in years (17-21, 22-30, 31-40, 41-55, 56-64, 65-74, 75-84, 85+), gender (male, female, rather not say), ethnicity (Asian, Black, White, Other, rather not say), use of

hospital transport (yes, no) and haemodialysis location (in-hospital, in-satellite, at home). Clinically relevant centre-level variables were selected. As the range of possible values differs between the constituent countries of the UK, for comparability, deciles for the Index of Multiple Deprivation were calculated. The strength of association between mean PREM score and prevalent mortality measures was assessed. These were the percentage of patients alive at the end of 2016 who died during 2017 (prevalent 1 year mortality unadjusted) and prevalent 1 year mortality adjusted to age 60 years”.

The Results paragraph two now includes the sentence: “No association was found between mean PREM score and prevalent mortality, either unadjusted ($r = 0.0482$), or age-adjusted ($r = -0.0500$)”.

Descriptive statistics have been added to table 2, which has also been amended to reduce the number of decimal places used to increase clarity.

Reviewer 2

As acknowledged by the authors, this study does leave the reader asking more questions than it answered. The authors could add to their discussion by comparing it to other related literature looking at the patient experience and patient/centre characteristics (e.g. Dad et al. 2018 BMC Nephrology)

We have extended the discussion of the literature in both the introduction and discussion, to improve the context of the literature throughout the paper – as detailed below.

Introduction p.4

Whether patient experience is mainly determined by patient characteristics (Van der Veer, Dad) or by those of the centre remains unknown (Brady).

Discussion p 14

Para 2

In previous studies, variation in ratings for self-reported patient experience amongst older patients, those experiencing shortened treatments ([Dad et al]) and some BAME populations ([Brady et al]) have been described.

Para 3

“In keeping with this Brady et al have reported variation in mean patient experience scores between groups of US facilities with different characteristics e.g., size of facility, hospital-based vs free-standing, for-profit vs non-profit. Differences were relatively small, yet still outweighed those attributed to patient characteristics”.

Tables 3 and 4 need footnotes explaining the abbreviations

We have added explanations into the tables or legends as appropriate

IMD = Index of Multiple Deprivation and
RRT = Renal Replacement Therapy

Other amendments

When the paper was drafted, the Kidney PREM was known as CKD PREM, which is now out-dated ‘branding’. Hence, all references to CKD-PREM in the text have been changed to Kidney PREM.

The ordering of the ethnicity variables in both text and tables is now alphabetical.

VERSION 2 – REVIEW

REVIEWER	Karine Manera The University of Sydney
REVIEW RETURNED	30-Jan-2021
GENERAL COMMENTS	We thank the author for their revisions, which have improved the manuscript. No further comments.